# A Comparative Analysis of the Fecal Bacterial Communities of Light and Heavy Finishing Barrows Raised in a Commercial Swine Production Environment

**DOI:** 10.3390/pathogens12050738

**Published:** 2023-05-20

**Authors:** Emily C. Fowler, Ryan S. Samuel, Benoit St-Pierre

**Affiliations:** Department of Animal Science, South Dakota State University, Animal Science Complex, Box 2170, Brookings, SD 57007, USA; emily.fowler@sdstate.edu

**Keywords:** 16S rRNA, microbiome, fecal, bacterial composition, barrow, swine, body weight

## Abstract

For commercial swine producers, the natural variation in body weight amongst pigs in a herd presents a challenge in meeting the standards of meat processors who incentivize target carcass weights by offering more favorable purchase prices. Body weight variation in a swine herd is evident as early as birth, and it is typically maintained throughout the entire production cycle. Amongst the various factors that can affect growth performance, the gut microbiome has emerged as an important factor that can affect efficiency, as it contributes to vital functions such as providing assimilable nutrients from feed ingredients that are inedible to the host, as well as resistance to infection by a pathogen. In this context, the objective of the study described in this report was to compare the fecal microbiomes of light and heavy barrows (castrated male finishing pigs) that were part of the same research herd that was raised under commercial conditions. Using high-throughput sequencing of amplicons generated from the V1-V3 regions of the 16S rRNA gene, two abundant candidate bacterial species identified as operational taxonomic units (OTUs), Ssd-1085 and Ssd-1144, were found to be in higher abundance in the light barrows group. Ssd-1085 was predicted to be a potential strain of *Clostridium jeddahitimonense*, a bacterial species capable of utilizing tagatose, a monosaccharide known to act as a prebiotic that can enhance the proliferation of beneficial microorganisms while inhibiting the growth of bacterial pathogens. OTU Ssd-1144 was identified as a candidate strain of *C. beijerinckii*, which would be expected to function as a starch utilizing symbiont in the swine gut. While it remains to be determined why putative strains of these beneficial bacterial species would be in higher abundance in lower weight pigs, their overall high levels in finishing pigs could be the result of including ingredients such as corn and soybean-based products in swine diets. Another contribution from this study was the determination that these two OTUs, along with five others that were also abundant in the fecal bacterial communities of the barrows that were analyzed, had been previously identified in weaned pigs, suggesting that these OTUs can become established as early as the nursery phase.

## 1. Introduction

The pork industry is an important part of the agriculture sector in the US, contributing over $57 billion to the American economy in 2021 [1]. As the aim of commercial pork producers is to provide high quality meat products to consumers at reasonable prices, a significant challenge to this goal is the natural variability in pig body size as animals grow from birth to market weight. Indeed, it is important for producers to sell animals at an ideal body weight, since meat processors purchase market hogs according to the carcass weight of each pig, with purchase prices incentivizing target carcass weights.

Variation in body weight within a swine herd is a consequence of a number of factors that can each influence individual growth, such as birth weight, overall health, as well as the level of intake during all production phases (nursing, nursery, growing and finishing) [2,3,4,5,6]. While growth rate can be impacted at any stage of production, variation in body weight generally increases with age. The factors that influence growth rate can be divided into two main categories: factors that affect the efficiency of nutrient acquisition from feed, and factors that affect the efficiency of nutrient utilization [7]. Due to its role in digestion and absorption of nutrients, the gastrointestinal tract is a critical system directly impacting the efficiency of conversion of feedstuffs into animal growth. In addition to its role in nutrition, the gut is also responsible for other functions that affect animal performance, such as immunity, and acting as a selective protective barrier to prevent the proliferation of pathogens and their invasion of the inner tissues [8]. In addition to host cells, the gastrointestinal tract also includes communities of microorganisms that benefit their host by contributing to both the nutrition and disease resistance functions of the gut [9,10]. Indeed, in a healthy gut environment, symbiotic microorganisms metabolize components of feed that are inedible by the host into short chain fatty acids [11], while also mitigating the risk of infection through competition with pathogens, production of antimicrobial compounds, and interactions with the immune system [12]. Thus, as the gut microbiome affects efficiency in animal production [13,14], alterations in its functions can potentially contribute to variation in body weight amongst individuals in a commercial herd.

While a number of published studies have reported on different factors that can modulate gut microbial composition in finishing pigs, such as feedstuffs [15,16,17,18,19,20,21,22,23], stage of development [24,25], sex [26,27], breed [28], adiposity [29], stress [30] or feed efficiency [31], limited information is available on the potential differences in microbial composition between light and heavy individuals within a swine herd. As barrows (castrated male finishing pigs) tend to grow faster than gilts, they typically represent a larger proportion of the pigs that are first marketed [6]; thus, insights that could help in improving their performance are of great interest to the swine industry. In this context, the study presented in this report compared the fecal bacterial composition of light and heavy finishing barrows of the same age that were raised at a research facility managed as a commercial wean-to-finish barn. In addition to the identification of candidate bacterial species that differed in abundance between the two weight classes, the analysis from this study also revealed that the majority of the most abundant bacteria in finishing barrows correspond to currently uncharacterized or unknown bacterial species.

## 2. Results

### 2.1. Animal Performance

During the first three weeks of the finishing phase for a herd of 1097 pigs at a research facility managed as a commercial wean-to-finish barn, pigs in barrow-only pens showed higher average body weights compared to pigs in gilt-only or mixed-sex pens (*p* < 0.05; Table 1). Once marketing of the heaviest animals was implemented at day 21, no differences in average body weight according to sex were observed amongst the pigs remaining at the facility (*p* > 0.05).

### 2.2. Taxonomic Composition Analysis of Fecal Bacterial Communities

To gain insight on the potential effect of the gut microbiome on different weight classes within a swine herd, a comparison of the fecal bacterial composition between ‘light’ (n = 21; weight range of 113–129 kg) and ‘heavy’ (n = 23; weight range of 137–156 kg) castrated male finishing pigs (barrows) was performed (Table 2; Appendix A). To this end, a combined total of 757,687 high quality sequence reads from the V1-V3 region of the 16S rRNA gene were generated from all samples (Appendix A). A taxonomy-based composition analysis identified Bacillota (formerly Firmicutes) as the most abundant phylum, with Clostridiaceae 1, Carnobacteriaceae, Ruminococcaceae and Peptostreptococcaceae as its most highly represented families (Table 3). Other abundant phyla identified included Planctomycetota, Bacteroidota (formerly Bacteroidetes) and Spirochaeta. Differences in abundance between the light and the heavy weight classes for the taxonomic groups analyzed were not found to be significant (*p* > 0.05).

### 2.3. Analysis of Alpha and Beta Diversity of Fecal Bacterial Communities

To gain further insight, an analysis based on OTU composition was performed. An alpha diversity analysis revealed that the Ace index was higher in the fecal communities of heavy pigs compared to light pigs (*p* < 0.05), while no differences were found for ‘observed OTUs’, Chao, Shannon or Simpson indices (Table 4). Of the 9066 OTUs identified in this study (Appendix A), 5050 OTUs were shared between the light and the heavy groups. These common OTUs represented 99.24% and 98.97% of sequences generated from the fecal samples collected from the light and heavy groups, respectively. While 1736 OTUs were found to be unique to the light group, and 2280 OTUs were only identified in the heavy group, these represented only a small percentage of the dataset (0.76% and 1.03% of sequence reads, respectively). The high degree of overlap between the two sets of samples was well illustrated using a beta diversity analysis, as there was no distinguishing clustering of samples according to weight groups (Figure 1).

### 2.4. OTU Composition Analysis of Fecal Bacterial Communities

Further analyses were conducted on individual OTUs with the goal of identifying potential bacterial species that were differentially represented between the light and heavy barrows. Amongst the most abundant OTUs, defined as representing at least 1.0% of sequences on average in at least one group of samples, two OTUs, Ssd-1085 and Ssd-1144, were found in higher abundance in barrows from the light group (*p* < 0.05). Notably, both OTUs showed a high nucleotide sequence identity to their respective valid relative (i.e., >97.8%), suggesting that they may represent strains of these species, while the other abundant OTUs were likely to correspond to novel bacterial species due to their limited sequence identity to their closest valid relative (80.91–93.42%). Fifteen other OTUs that were present in lower abundance were found to be differentially represented between the two groups; these included four OTUs that were higher in the light group, and eleven OTUs that were higher in the heavy group (Figure 2; Appendix A). Intriguingly, these fifteen differentially represented OTUs would be expected to correspond to unknown bacterial species based on their nucleotide sequence identity to their respective closest relative (80.32–96.20%).

## 3. Discussion

Meat processors purchase market hogs according to the carcass weight of each animal, with target carcass weights incentivized by offering more favorable prices. Since it is important for producers to sell animals at an ideal body weight, the intra-pen variation in body weight of individual pigs becomes important when pigs are marketed [5]. Variation in body weight within each litter of pigs exists at birth, and weight variation is typically maintained until pigs are marketed [2,4]. Body weight variation generally increases with age due to differences in the growth rate of individual animals during lactation [3], as well as after weaning [5,6]. Since barrows generally grow faster than gilts and are thus expected to represent a larger proportion of the pigs first marketed [6], information about the differences between light and heavy finishing barrows of the same age is of great interest to the swine industry.

In light of the contribution of the gut microbiome to the nutrition of the host and resistance to pathogens, the aim of the study presented in this report was to compare the fecal bacterial composition of light and heavy barrows that were raised under the same conditions, in order to identify candidate bacterial species that varied in abundance between the two weight classes. Ssd-1085 and Ssd-1144, two of the most abundant OTUs in fecal communities of finishing barrows, were found to be more highly represented in animals of the lower weight group. Based on nucleotide sequence identity analysis, Ssd-1085 was predicted to be a strain of *Clostridium jeddahitimonense* a bacterial species originally isolated from human feces [33]. It has been described as a Gram positive, endospore forming, non-motile bacilli that is a strict anaerobe, with optimal growth at 37 °C. Intriguingly, *C. jeddahitimonense* has been found to metabolize D-tagatose, but it is unable to utilize common monosaccharides such as glucose, fructose, galactose, ribose, xylose or arabinose [33]. Most tagatose (~80%) is not absorbed by the host, due to the limited capacity of animal cells to metabolize this compound. Instead, tagatose acts as a prebiotic by enhancing the proliferation of beneficial microorganisms [34], and as an inhibitor of growth for bacterial pathogens such as *Listeria monocytogene* and *Salmonella typhimurium* [35]. The ability to utilize tagatose appears to be limited to certain types of gut microorganisms, a group that includes *Enterococcus faecalis* [34]. Notably, tagatose has been reported as an abundant metabolite in soybean, with a potential role in response to abiotic stress [36]. In contrast, OTU Ssd-1144 was predicted to be a strain of *C. beijerinckii*, an anaerobe capable of hydrolyzing starch and fermenting glucose [37,38], which are metabolic functions that are likely more common amongst gut microbial species than the utilization of tagatose. While it remains to be determined why putative strains of *C. beijerinckii* and *C. jeddahitimonense* would be in higher abundance in lower weight pigs, their overall high levels in finishing pigs could be the result of including ingredients such as corn and soybean-based products, respectively, in swine diets.

Other insights provided by this report include the identification of currently uncharacterized gut bacterial species from swine raised in a commercial setting. In this category, Ssd-1095 could be considered the most intriguing, as it was not only the most abundant OTU identified, with a range of 8.66–43.05% across all samples, but also because it represented approximately 99% of sequences affiliated to Planctomycetota in both light and heavy barrows. Since it could only be assigned as an unclassified Planctomycetacia, this OTU likely belonged to a currently undefined order within the phylum Planctomycetota. This is consistent with its very low sequence identity to its closest valid relative (*Lignipirellula cremea.* 80.91%). In addition, this OTU, along with six other abundant OTUs reported in Table 5, has been reported in nursery pigs from previous studies by our group [39,40,41,42]. As their representation in nursery pigs was lower than observed in the finishing pigs analyzed in this current study, these OTUs may represent bacterial species that are established after weaning, then later become more prominent as pigs continue to grow. As we strive to further elucidate the specialized functions of gut symbiotic microorganisms at the species level, it will be of equal interest not only to determine the metabolic potential of uncharacterized species, but also to determine what conditions or feed ingredients promote the growth of these unknown members of the gut microbiota.

## 4. Conclusions

The variation in body size that is naturally occurring in commercial swine herds presents a challenge to producers who aim to meet the target standards for carcass weights set by meat processors. As a contribution towards gaining more insight on the potential effect of the gut microbiome on weight variation in pigs, this report described differences in the representation of candidate bacterial species, identified as OTUs, which were found between heavy and light weight market pigs from the same herd. While it remains to be determined whether they can impact body weight or whether they are a secondary effect to their host’s physiology, these OTUs provide an intriguing possibility that they could be targeted as part of intervention strategies to help mitigate the extent of body weight variation in commercial herds. Notably, two of these OTUs were amongst the most abundant in the fecal bacterial communities of pigs, with each predicted to utilize different substrates that are present in feedstuffs. As these OTUs have also been found in nursery pigs during previous studies by our group, albeit at much lower levels, this presents the possible development of dietary interventions in post-weaned pigs to modulate their gut microbiome during the growing and finishing phases to optimize their abundance. Alternatively, their higher representation in lower weight pigs may serve as indicators of physiological or immunological inefficiencies, warranting future research in elucidating their function in the swine gut.

## 5. Materials and Methods

### 5.1. Animals and Sample Collection

The trial described in this report was conducted at the South Dakota State University (SDSU) Off-Site Wean-to-Finish Barn, with all procedures approved by the SDSU Institutional Animal Care and Use Committee before the start of the study (2005-026E). The facility was populated with weaned pigs (Babcock genetic line; ~5.4–6.8 kg/pig) purchased from a single producer. The weaned pigs were mixed from multiple litters as a result of transportation and random assignment to pens upon their arrival. Once at the grower stage, (~50 kg; ~80 days of age), all pigs (n = 1097) were divided into three different sex-based categories (barrows, gilts or mixed sex) that were randomly divided into 45 pens (24–25 pigs/pen) distributed between two rooms. The 15 pens of each sex were approximately equal between the rooms. The standard pen dimensions (3.1 m × 6.9 m) provided pen space of 0.88 m^2^/pig.

Diets with corn and soybean meal as the major feedstuffs were provided as grower and finisher phases during the trial. Standard grower and finisher diets were formulated to meet nutrient requirements according to the NRC (2012) guidelines [43] (Appendix A). Grower pigs were provided diets formulated to provide 0.81%, 0.72%, and 0.64% digestible lysine from 50 to 64 kg, 64 to 82 kg, and 82 to 100 kg, respectively. Finishing pigs were provided diets formulated to provide 0.57% digestible lysine and 0.48% digestible lysine from 100 to 120 kg and from 120 to 160 kg, respectively. Once the average pig weight reached 135 kg, the heaviest pigs from each pen were marketed over the following three weeks.

The selection of heavy and light pigs that were analyzed for this study took place one day before the first pigs were sent to the processing plant. Individuals from the six pens with only barrows (castrated male finishing pigs) that had the heaviest average body weight (Appendix A) were individually weighed; barrows weighing less than 125 kg were assigned to the ‘light’ group (n = 21) while barrows weighing more than 139 kg (n = 23) were assigned to the ‘heavy’ group. Fresh fecal samples were subsequently collected on the same day using rectal stimulation of the selected pigs, and these were stored frozen at −20 °C until they were processed for microbial genomic extraction.

### 5.2. Isolation of Microbial Genomic DNA and Sequencing of 16S rRNA Gene Amplicons

Microbial genomic DNA was extracted from individual samples using a bead-beating plus column approach as previously described [44], which included use of the QIAamp DNA Mini Kit (Qiagen, Hilden, Germany). The V1-V3 regions of the bacterial 16S rRNA gene were targeted by PCR using the universal forward 27F-5’AGAGTTTGATCMTGCTCAG [45] and reverse 519R-5’GWATTACCGCGCGCGCTG [46] primers. Purified microbial genomic DNA samples were submitted to Molecular Research DNA (MRDNA, Shallowater, TX, USA) for V1-V3 amplification and amplicon sequencing with the Illumina MiSeq 2 × 300 platform to generate overlapping paired-end reads.

### 5.3. Bacterial Composition Analyses

Sequence data were processed using a combination of custom-written Perl scripts and publicly available software. Sequences from merged overlapping paired-end reads corresponding to V1-V3 amplicons generated from the 16S rRNA bacterial gene were first screened to meet the following criteria: presence of both intact 27F and 519R primer sequences, a minimal average Phred quality score of Q33, and length between 400 and 580 nt [39]. Following quality filtering, amplicon sequences were aligned, then clustered into operational taxonomic units (OTUs) using a sequence dissimilarity cutoff of 4%. Based on previously published reports [47,48], this threshold is more suitable for the V1-V3 region than the 3% cutoff that is typically used indiscriminately for clustering of 16S rRNA sequence data, regardless of the variable regions targeted for analysis. Following OTU clustering, three independent approaches were used to identify artifacts [39]. First, OTUs were screened for chimeric sequences using the ‘chimera.slayer’ [49] and ‘chimera.uchime’ [50] commands from the MOTHUR (v.1.36.1) open-source software package [51]. Secondly, the 5′ and 3′ ends of OTUs were evaluated using a database alignment search-based approach; when compared to their closest match of equal or longer sequence length from the NCBI ‘nt’ database, as determined by BLAST [52], OTUs with more than five nucleotides missing from the 5′ or 3′ end of their respective alignments were designated as artifacts. Finally, OTUs with only one or two assigned reads were subjected to an additional screening, wherein only sequences with a perfect or near-perfect match (maximum 1% of dissimilar nucleotides) to a sequence in the NCBI ‘nt’ database were kept for analysis. All OTUs and their assigned reads that were flagged during these screens were subsequently removed from further analyses. The resulting curated OTUs were then analyzed for taxonomic assignment using two strategies. For all OTUs, phylum and family-level affiliations were determined using RDP Classifier [53]. The closest valid relatives for the most abundant OTUs were identified by searches with BLAST against the ‘refseq_rna’ database [52].

Using the MOTHUR (v.1.44.1) open-source software package [51], the alpha diversity indices ‘Observed OTUs’, ‘Chao’, ‘Ace’ and ‘Shannon’ were determined using the ‘summary.single’ command. For beta diversity analysis in MOTHUR (v.1.44.1), Bray–Curtis distances were first calculated using ‘summary.shared’, followed by ‘pcoa’ for Principal Coordinate Analysis (PCoA). To perform alpha and beta diversity analyses, curated sequence datasets were first rarefied using custom Perl scripts to 8474 sequences, which was the lowest read count amongst the samples analyzed (Appendix A). The plot for PCoA was generated using Tableau Visualization Software (Version 2020.4, https://www.tableau.com/products/new-features; accessed on 7 April 2023).

### 5.4. Statistical Analyses

Statistical analysis for body weights were performed using the MIXED procedure of SAS with sex as the class variable. Statistical analysis of alpha indices (parametric data) was performed with a *t*-test using the online platform GraphPad (https://www.graphpad.com/quickcalcs/ttest1.cfm; accessed on 7 April 2023). For the statistical analysis of taxonomic groups and most abundant OTUs (non parametric data), the Wilcoxon rank-sum test was performed in ‘R’ (Version 3.6.0). For all statistical tests described above, a threshold of *p* ≤ 0.05 was considered significant. Analysis by LDA effect size (LEfSe) [32] was performed using the publicly available online platform (http://huttenhower.sph.harvard.edu/galaxy/; accessed on 10 April 2023).

### 5.5. Next Generation Sequencing Data Accessibility

Raw sequence data are available from the NCBI Sequence Read Archive under Bioproject PRJNA963964.

## Figures and Tables

**Figure 1 pathogens-12-00738-f001:**
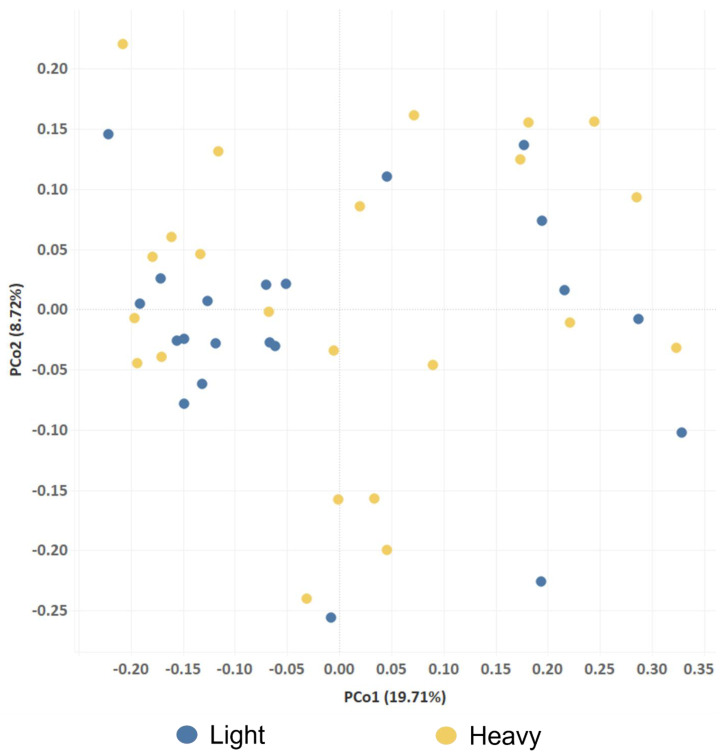
Comparison of fecal bacterial communities between light and heavy barrows using principal coordinate analysis (PCoA). PCoA was performed using a Bray-Curtis distance matrix. The x and y axes correspond to principal components 1 (PCo1) and 2 (PCo2).

**Figure 2 pathogens-12-00738-f002:**
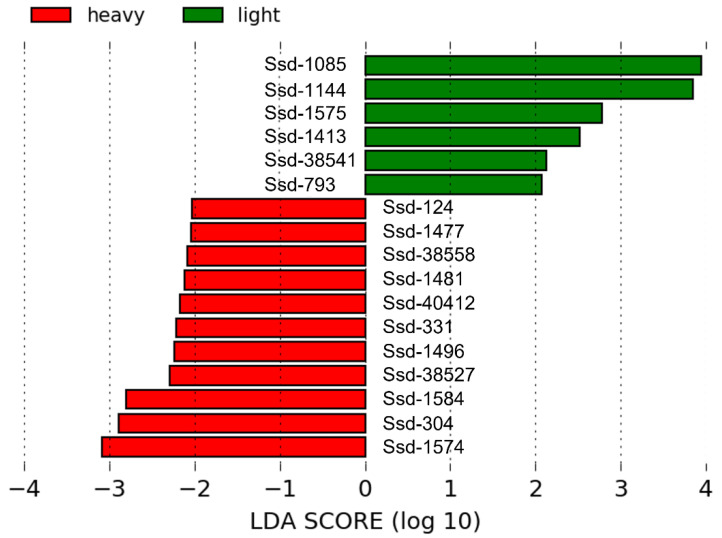
OTUs identified by LDA effect size (LEfSe) [32] as characteristic of heavy and light pigs, respectively. The abundance and taxonomic affiliation of these OTUs can be found in Appendix A.

**Table 1 pathogens-12-00738-t001:** Weekly mean body weight ^1^ (kg) of finishing pigs from 105.2 ± 0.5 kg to heavy market weight (>135 kg) that were housed in pens of either only barrows or gilts, or had a mixed sex population.

	Pen Composition				
Week	Barrow	Gilt	Mixed	SEM	*p*-Value
0	108.0 ^a^	103.7 ^b^	103.9 ^b^	0.5	<0.01
1	115.5 ^a^	110.6 ^b^	111.2 ^b^	0.5	<0.01
2	122.5 ^a^	116.8 ^b^	117.8 ^b^	0.5	<0.01
3	129.0 ^a^	123.8 ^b^	124.5 ^b^	0.5	<0.01
4	130.4	130.0	130.9	0.5	0.72
5	133.6	133.7	135.1	0.6	0.57
6	-	137.9	139.0	0.6	0.37

^1^ Values were determined by dividing the pen weight by the number of individuals in each respective pen. Different superscripts in the same row indicate that groups were statistically different (*p* < 0.05).

**Table 2 pathogens-12-00738-t002:** Mean body weight at the time of sampling for the two groups of barrows analyzed in this study.

Group	Mean BW (kg ± SEM)	Range BW (kg)	n
Light	119.5 ^a^ ± 0.8	112.7–124.1	21
Heavy	145.9 ^b^ ± 1.1	139.1–160.0	23
Pen mean	133.0 ± 1.1	129.6–137.5	6

Different superscripts in the same column indicate that groups were statistically different (threshold was *p* < 0.05; calculated *p* value < 0.0001).

**Table 3 pathogens-12-00738-t003:** Mean relative abundance ^1^ (%) of main bacterial phyla and families identified in light and heavy groups.

Taxonomic Affiliation	Light	Heavy
**Bacillota**	**43.86 ± 2.61**	**45.60 ± 2.00**
Clostridiaceae 1	13.43 ± 1.36	10.79 ± 1.11
Carnobacteriaceae	5.51 ± 1.87	8.70 ± 2.34
Ruminococcaceae	4.23 ± 0.22	5.08 ± 0.37
Peptostreptococcaceae	2.58 ± 0.27	2.26 ± 0.22
Unclassified Clostridiales ^&^	7.45 ± 0.70	8.85 ± 0.63
Other Bacillota ^&^	10.66 ± 1.31	9.91 ± 0.84
**Planctomycetota**	**24.75 ± 2.16**	**24.20 ± 2.16**
Unclassified Planctomycetacia ^&^	24.68 ± 2.16	24.13 ± 2.15
Other Planctomycetota ^&^	0.07 ± 0.01	0.08 ± 0.01
**Bacteroidota**	**14.79 ± 1.20**	**16.27 ± 1.05**
Prevotellaceae	2.92 ± 0.40	3.75 ± 0.44
Unclassified Bacteroidales ^&^	8.91 ± 0.90	9.23 ± 0.66
Other Bacteroidota ^&^	2.96 ± 0.31	3.30 ± 0.33
**Spirochaetota**	**8.37 ± 1.12**	**6.80 ± 0.77**
Spirochaetaceae	8.29 ± 1.12	6.73 ± 0.76
Other Spirochaetota ^&^	0.08 ± 0.01	0.07 ± 0.01
**Other Bacteria ** ^&$^	**8.23 ± 2.39**	**7.12 ± 0.97**

^1^ Mean relative abundance of taxonomic groups is presented as a percentage (%) of the total number of analyzed reads per sample. Phylum-level groups are highlighted in bold, followed by their respective family-level groups. ^&^ Statistical test not performed because of group heterogeneity. ^$^ ‘Other bacteria’ included sequences affiliated to the phyla Pseudomonadota, Mycoplasmatota, Fusobacteriota, Synergistota, Candidatus Saccharibacteria, Lentisphaerota, Actinomycetota, Candidate division WPS-1, Chloroflexota, Fibrobacterota, Nitrospirota, Candidatus Cloacimonadota, Campilobacterota, Acidobacteriota, and Verrucomicrobiota, as well as unclassified bacteria.

**Table 4 pathogens-12-00738-t004:** Mean of alpha diversity indices for the light and heavy groups of barrows analyzed in this study.

Index	Light (±SEM)	Heavy (±SEM)	*p*-Value
Observed OTUs	811.4 ± 18.5	853.1 ± 18.0	0.1133
Chao	1750.8 ± 38.6	1880.9 ± 51.3	0.0521
Ace	2606.0 ^a^ ± 71.2	2866.7 ^b^ ± 86.0	0.0259
Shannon	3.89 ± 0.08	4.04 ± 0.08	0.2043
Simpson	0.11 ± 0.01	0.10 ± 0.01	0.5203

Different superscripts in the same row indicate that groups are statistically different (*p* < 0.05).

**Table 5 pathogens-12-00738-t005:** Mean relative abundance ^$^ (%) and taxonomic affiliation of the most abundant bacterial OTUs identified in this study.

OTUs	Light	Heavy	Closest Valid Relative (id% *)
**Bacillota**			
Ssd-1085 ^1,3,4^	5.88 ^a^ ± 0.67	4.16 ^b^ ± 0.37	*Clostridium jeddahitimonense* (98.59%)
Ssd-1398	3.45 ± 1.27	5.25 ± 1.65	*Carnobacterium funditum* (93.22%)
Ssd-1144 ^1,2,3,4^	3.62 ^a^ ± 0.44	2.22 ^b^ ± 0.27	*Clostridium beijerinckii* (97.80%)
Ssd-0675 ^1,2,3^	2.19 ± 0.75	1.79 ± 0.48	*Christensenella massiliensis* (84.54%)
Ssd-1566	1.12 ± 0.52	1.76 ± 0.69	*Carnobacterium funditum* (93.42%)
Ssd-1079 ^1,3,4^	1.44 ± 0.29	1.62 ± 0.26	*Mahella australiensi* (83.01%)
**Bacteroidota**			
Ssd-1048 ^1,3^	3.09 ± 0.59	3.07 ± 0.36	*Caecibacteroides pullorum* (86.93%)
**Planctomycetota**			
Ssd-1095 ^1,2,3,4^	24.52 ± 2.14	23.97 ± 2.09	*Lignipirellula cremea* (80.91%)
**Spirochaetota**			
Ssd-1115 ^1,4^	4.58 ± 0.79	3.81 ± 0.45	*Treponema peruense* (84.62%)
Ssd-1399	3.16 ± 0.72	2.34 ± 0.53	*Treponema bryantii* (89.53%)

^$^ Mean relative abundance of OTUs is presented as a percentage (%) of the total number of analyzed reads per sample. Different letter superscripts in the same row indicate that groups were statistically different (*p* < 0.05) by the Wilcoxon rank sum test. * Nucleotide sequence identity (%) between each OTU and its corresponding closest valid relative. Number superscripts indicate that the corresponding OTUs were identified in previously published studies: ^1^ Poudel et al., 2022 [39]; ^2^ Poudel et al., 2020 [40]; ^3^ Zeamer et al., 2021 [42]; ^4^ Fresno Rueda et al., 2021 [41].

## Data Availability

Raw sequence data are available from the NCBI Sequence Read Archive under Bioproject PRJNA963964.

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
