# Peer review of "A Comparative Analysis of the Fecal Bacterial Communities of Light and Heavy Finishing Barrows Raised in a Commercial Swine Production Environment"

_pathogens, 2023, doi:10.3390/pathogens12050738_

Round 1

Reviewer 1 Report

The diverse roles of intestinal bacteria, the microbiome, now is well established not only in human but also animal studies, veterinary and agricultural.  The present is a very focused study to determine if differences exist which might explain the ability of pigs, in particular castrated male barrows, to grow quicker, making them ready for use earlier.  The authors provide a very good background which cites the most cited literature on the pig microbiome.   The manuscript presentation is clear, concise, thoughtful, and conservative.   They present data in a manner to determine what speculations might be made, thus conservative.  There is much work to be done such as to determine the mechanisms of action and even identity of many of these bacterial species of the pig microbiome, but this is an important start that makes a contribution. 

Author Response

Response to reviewer comments (pathogens-2403375)

Reviewer 1

The diverse roles of intestinal bacteria, the microbiome, now is well established not only in human but also animal studies, veterinary and agricultural.  The present is a very focused study to determine if differences exist which might explain the ability of pigs, in particular castrated male barrows, to grow quicker, making them ready for use earlier.  The authors provide a very good background which cites the most cited literature on the pig microbiome.   The manuscript presentation is clear, concise, thoughtful, and conservative.   They present data in a manner to determine what speculations might be made, thus conservative.  There is much work to be done such as to determine the mechanisms of action and even identity of many of these bacterial species of the pig microbiome, but this is an important start that makes a contribution.

AUTHOR RESPONSE: Thank you for your review, and we greatly appreciate your positive assessment and feedback.

Reviewer 2 Report

Comments to the Authors of manuscript number: pathogens-2403375 entitled “A comparative analysis of the fecal bacterial communities of light and heavy finishing barrows raised in a commercial swine production environment”.

It is a very interesting study, but the study design is described very poorly. It should be corrected and presented very clear to understand the issue presented. Authors mentioned all factors influencing the difference in the weight between animals, thus They should described everything in proper manner starting from producers, litters and other factors that could have the influence of body weight. It will give the proper look on the results.

1. L 7 – add the name of country

2. Introduction is comprehensive and presents the issue very well

3. part 4.1. – How animals were chosen? Was it the own animal production? How many litters were included into the study? What was the average weight in these litters? If animals were bought, from how many producers? How animals were distributed in mixed pens? How many barrows and gilts was in such pen? Generally, the study design is poorly described.

4. Table presenting diets of each stage should be provided

5. If animals at the start weighed app.50 kg, why they were fed diet for animals of 100 to 160kg?

6. at what time animals were chosen to final part of the study, when the lightest and the heaviest pigs were chosen?

Author Response

Response to reviewer comments (pathogens-2403375)

Reviewer 2

Comments to the Authors of manuscript number: pathogens-2403375 entitled “A comparative analysis of the fecal bacterial communities of light and heavy finishing barrows raised in a commercial swine production environment”.

It is a very interesting study, but the study design is described very poorly. It should be corrected and presented very clear to understand the issue presented. Authors mentioned all factors influencing the difference in the weight between animals, thus They should described everything in proper manner starting from producers, litters and other factors that could have the influence of body weight. It will give the proper look on the results.

AUTHOR RESPONSE: Thank you for your review and positive feedback. We hope that we have addressed your comments and concerns to your satisfaction (see below). Please note that, to facilitate your review of the revised manuscript, text boxes have been added to mark the location of revisions associated with each comment, and the revised text has been highlighted.

We apologize for any confusion regarding the description of the study. The comparison between light and heavy pigs described in this manuscript was not the aim of a specific trial designed with this goal in mind, but rather an opportunity to compare the fecal bacterial communities of light and heavy pigs raised in the same facility and fed the same diet. The range in BW in commercial herds is a common challenge in the industry, and we adopted an opportunistic approach to gain further insight into the potential involvement or effect of the gut microbiome. While we acknowledge that this approach may be unorthodox, having a herd of ~ 1000 pigs at our disposal to select from allowed us to maximize the range in body weight available for an opportunistic study. This was a practical way to maximize the use of available resources to gain more insight on the gut microbiome of swine. If we had used a more conventional approach, designing a study to compare extreme weight classes, we are not confident that we could have secured funding for a trial with 1,00 pigs, and designing a trial with fewer pigs would likely not have given us a very broad range of body weights to choose from.

R2-C1

  1. L 7 – add the name of country

AUTHOR RESPONSE:  ‘U.S.A.’ was added to the address as requested.

R2-C2

  1. Introduction is comprehensive and presents the issue very well

AUTHOR RESPONSE:  Thank you for your positive assessment and feedback.

R2-C3

  1. part 4.1. – How animals were chosen? Was it the own animal production? How many litters were included into the study? What was the average weight in these litters? If animals were bought, from how many producers? How animals were distributed in mixed pens? How many barrows and gilts was in such pen? Generally, the study design is poorly described.

 AUTHOR RESPONSE:  Thank you for bringing this to our attention:

The barn was populated with weaned piglets purchased from a single producer, and South Dakota State University was therefore the owner of the pigs.

There was no available information on litters that could address the questions above. Piglets from multiple litters were mixed as a result of transportation (transport from a separate facility by truck), and of the random assignment to pens once weaned pigs arrive at the barn. In addition, pigs were later sorted based on sex once they reached ~50 kg.

Similarly, information on average litter weight was not available – typically, weaned pigs arrive at the facility with a weight of 12-15 lbs.

There was no specific design for mixed pens. Pens that were exclusively populated with gilts or barrows were filled first, then mixed pens were randomly populated with pigs that were left over, aiming to have approximately 50% of each sex.

We have summarized this information into the following text which has been added to the revised manuscript.

Revised MS, lines 326 – 329: “The facility was populated with weaned pigs (Babcock genetic line; ~ 5.4 – 6.8 kg / pig) purchased from a single producer. The weaned pigs were mixed from multiple litters as a result of transportation and random assignment to pens upon their arrival.”

R2-C4

  1. Table presenting diets of each stage should be provided

    AUTHOR RESPONSE:  Thank you for bringing this to our attention. Diets were added as supplementary material (Supplementary Table 5), which was referenced on lines 337-338 of the revised manuscript.

R2-C5

  1. If animals at the start weighed app.50 kg, why they were fed diet for animals of 100 to 160kg?

AUTHOR RESPONSE:  We apologize for the confusion. The missing information below was added to the text.

Revised MS, lines 338 - 339: “Grower pigs were provided diets formulated to provide 0.81%, 0.72%, and 0.64% digestible lysine from 50 to 64 kg, 64 to 82 kg, and 82 to 100 kg, respectively.”

R2-C6

  1. at what time animals were chosen to final part of the study, when the lightest and the heaviest pigs were chosen?

  AUTHOR RESPONSE:  Individual animals were weighed one day before the first pigs were sent to the processing plant. On the same day, heavy and light weight barrows were identified, and fecal samples were collected. The text of the revised manuscript now reads:

Revised MS, lines 344 - 345: “The selection of heavy and light pigs that were analyzed for this study took place one day before the first pigs were sent to the processing plant.”

Revised MS, lines 349 - 350 “Fresh fecal samples were subsequently collected on the same day by rectal stimulation (…).”

Round 2

Reviewer 2 Report

I have no comments